# Clinical Features of COVID-19 in Pediatric Rheumatic Diseases: 2020–2022 Survey of the Pediatric Rheumatology Association of Japan

**DOI:** 10.3390/v15051205

**Published:** 2023-05-20

**Authors:** Hiroyuki Wakiguchi, Utako Kaneko, Satoshi Sato, Tomoyuki Imagawa, Hidehiko Narazaki, Takako Miyamae

**Affiliations:** 1Department of Pediatrics, Yamaguchi University Graduate School of Medicine, Ube 755-8505, Japan; 2Department of Pediatrics, Niigata University Graduate School of Medical and Dental Sciences, Niigata 951-8510, Japan; utako-k@med.niigata-u.ac.jp; 3Department of Infectious Diseases and Immunology, Saitama Children’s Medical Center, Saitama 330-8777, Japan; sato.satoshi@saitama-pho.jp; 4Department of Infection and Immunology, Kanagawa Children’s Medical Center, Yokohama 232-0066, Japan; timagawa@kcmc.jp; 5Department of Pediatrics, Nippon Medical School, Tokyo 113-8602, Japan; nara@nms.ac.jp; 6Department of Pediatric Rheumatology, Institute of Rheumatology, Tokyo Women’s Medical University Hospital, Tokyo 162-8666, Japan; tmiyamae@twmu.ac.jp

**Keywords:** children, juvenile idiopathic arthritis, omicron, questionnaire, SARS-CoV-2, severity, systemic lupus erythematosus

## Abstract

Coronavirus disease 2019 (COVID-19) in children can be compounded by concurrent diseases and immunosuppressants. For the first time, we aimed to report the clinical features of concurrent COVID-19 and pediatric rheumatic disease (PRD) in Japan. Pediatric Rheumatology Association of Japan members were surveyed between 1 April 2020 and 31 August 2022. Outcome measurements included the clinical features of concurrent PRD and COVID-19. Questionnaire responses were obtained from 38 hospitals. Thirty-one hospitals (82%) had children with PRD and COVID-19. The female-to-male ratio in these children (*n* = 156) was 7:3, with half aged 11–15 years. The highest proportion of children with PRD and COVID-19 was accounted for by juvenile idiopathic arthritis (52%), followed by systemic lupus erythematosus (24%), juvenile dermatomyositis (5%), scleroderma (4%), and Takayasu arteritis (3%). Of children with PRD, a significant majority (97%) were found to be asymptomatic (10%) or presented with mild symptoms (87%) of the COVID-19 infection. No severe cases or deaths were observed. Regarding the use of glucocorticoids, immunosuppressants, or biologics for PRD treatment before COVID-19, no significant difference was found between asymptomatic/mild and moderate COVID-19 in children with PRD. Therefore, COVID-19 is not a threat to children with PRD in Japan.

## 1. Introduction

The coronavirus disease 2019 (COVID-19), caused by severe acute respiratory syndrome coronavirus-2 (SARS-CoV-2), has shown a remarkably high rate of spread since December 2019 [1]. According to the data reported as of August 2022 by the World Health Organization, more than 598.0 million and 6.4 million people worldwide have been infected and died, respectively [2]. From 1 August 2021 to 31 July 2022, COVID-19 was among the 10 leading causes of death in children and young people aged 0 to 19 years in the United States, ranking eighth among all causes of death, fifth in disease-related causes of death, and first in deaths caused by infectious or respiratory diseases [3]. COVID-19 deaths constituted 2% of all causes of death in this age group during this time frame. Initially, a low prevalence rate in children with COVID-19 was reported in various countries, and it was thought that children were unlikely to be affected [4,5,6]. Subsequently, it became clear that pediatric morbidity was underestimated. Some studies reported prevalence rates of up to approximately 22%, with half of the cases being asymptomatic [7,8]. Since the Delta variant epidemic, there has been a rapid spread of infection among children through schools, indoor sports facilities, and homes, resulting in a significant increase in pediatric cases and the number of hospitalizations worldwide, including in Japan [9,10,11]. The Omicron variant epidemic increased the number of children being hospitalized in many countries, but the hospitalization risk was reduced by a factor of three when compared with that of the Delta variant [12,13,14]. According to the national data collected by the Japanese government using a system called HER-SYS, from 2 September 2020 to 30 August 2022, which included the epidemic seasons of the Alpha, Delta, and Omicron variants, 2,231,776 children < 10 years and 2,296,905 teenagers contracted COVID-19 [15]. Of these, 139 children < 10 years and 64 teenagers contracted severe COVID-19. As a rule, severe cases are defined as meeting one of the following conditions: (1) being connected to a mechanical ventilator, (2) being on extracorporeal membrane oxygenation, or (3) being treated in the intensive care unit (or a similar facility) [15].

Pediatric COVID-19 infections are associated with specific sequela. Multisystem inflammatory syndrome in children (MIS-C) is a postinfectious complication of SARS-CoV-2 infection primarily affecting children [16]. Many patients with MIS-C had a documented history of COVID-19 within the two months before developing MIS-C during the Omicron variant epidemic [17]. In some patients, there is no time separation between the viral-induced symptoms and MIS-C. Moreover, MIS-C shares features with pediatric rheumatic disease (PRD) and cytokine storm syndrome, frequently requiring intensive care support. Although intravenous immunoglobulin and glucocorticoids are effective therapeutics for most patients with MIS-C, patients with refractory MIS-C are treated with various biologics. Understanding the clinical features and the role of inflammatory cytokines provides a rationale for using biologics in treating severe MIS-C [16].

Conversely, COVID-19 in children can be compounded by the presence of underlying diseases and the use of glucocorticoids, immunosuppressants, or biologics; therefore, further research in this area is required to gather comprehensive data [18]. Additionally, these drugs may modify the symptoms of COVID-19 [19]. Thus, this study aimed to investigate the current status of children with PRD who were infected with COVID-19 while using immunosuppressants. Here, we report the clinical features of COVID-19 in children with PRD in Japan for the first time.

## 2. Materials and Methods

### 2.1. Study Population and Data

An online survey was conducted among members (pediatricians [majority]/physicians [minority]) of the Pediatric Rheumatology Association of Japan (PRAJ) at 318 hospitals, including 58 PRD Center Facilities in Japan. The PRD Center Facilities are base hospitals throughout Japan, certified by the PRAJ for treating patients with PRD [20]. All PRAJ members affiliated with the PRD Center Facilities are pediatricians. We prepared 10 questions (Q1–10) for the questionnaire (Table 1). Q1 was used to confirm the presence or absence of patients with PRD (age < 18 years) with COVID-19 at each hospital in Japan. Q2–Q7 revealed the number of patients with PRD and COVID-19 in the hospital in Japan. Q8–Q10 asked about the practices of pediatricians/physicians who examined patients with PRD and COVID-19 in Japan.

In Q5, mild cases were defined as those that improved without the use of pharmacological agents for COVID-19 treatment; moderate cases were defined as those where pharmacological agents were used to treat COVID-19 but management in the intensive care unit was not required; and severe cases were defined as those that required intensive care unit management.

The survey period was from 1 April 2020 to 31 August 2022, which included the epidemic seasons of the Alpha, Delta, and Omicron variants, and the response period was from 22 September to 4 October 2022. All members of the PRAJ were notified by email on 20 September 2022, that the survey would be conducted. The survey was only distributed during the response period.

### 2.2. Outcome Measurements

Outcome measurements included the demographic characteristics and clinical features of COVID-19 in patients with PRD and the practices of pediatricians/physicians. Demographic characteristics included sex, age, and PRD type. Clinical features included the severity of COVID-19 and the relationship between pharmacological agents used for PRD treatment (glucocorticoids, immunosuppressants, or biologics) and COVID-19 severity. The practice of pediatricians/physicians for patients with PRD and COVID-19 included adjustments to pharmacological agents used to treat PRD (glucocorticoids, immunosuppressants, or biologics) during COVID-19 infection.

### 2.3. Statistical Analyses

For statistical analysis of the relationship between pharmacological agents used to treat PRD and COVID-19 severity, Fisher’s exact test was used to compare variables between the two groups. Statistical significance was set at *p* < 0.05. Statistical analyses were performed using JMP Pro version 16 (SAS Institute, Cary, NC, USA).

## 3. Results

### 3.1. Number of Hospitals That Had Children with PRD and COVID-19

Questionnaire responses were obtained from 38 hospitals (a response rate of 12% [38/318]), including 22 PRD Center Facilities (a response rate of 38% [22/58]) and 16 non-PRD Center Facilities (a response rate of 6% [16/260]) in Japan. The responding hospitals were evenly distributed, including the northernmost, eastern, central, western, and southernmost parts of Japan. In response to Q1, 31 hospitals (82% [31/38]), including 22 PRD Center Facilities (100% [22/22]) and nine non-PRD Center Facilities (56% [9/16]), confirmed having had children with PRD and COVID-19.

### 3.2. Number, Sex, Age, PRD Type, and COVID-19 Severity in Children with PRD and COVID-19

Table 2 presents the answers to Q2–Q5. In total, there were 165 children with COVID-19, but nine had Crohn’s disease without arthritis and were excluded from the analysis. In children with PRD and COVID-19 (*n* = 156), the female-to-male ratio was 7:3, with half of the children aged 11–15 years. The next most common age groups were 6–10 years (23%) and 16–17 years (21%). In children with PRD and COVID-19, the highest proportion was accounted for by juvenile idiopathic arthritis (JIA) (52%), followed by systemic lupus erythematosus (SLE) (24%), juvenile dermatomyositis (5%), scleroderma (4%), Takayasu arteritis (3%), chronic recurrent multifocal osteomyelitis (3%), and Sjögren’s syndrome (2%). Antiphospholipid antibody syndrome, Behçet’s disease, mixed connective tissue disease (MCTD), polyarteritis nodosa, arthritis associated with Crohn’s disease, juvenile polymyositis, overlap syndrome, tubulointerstitial nephritis and uveitis syndrome, and uveitis accounted for approximately 1%. Regarding the severity of COVID-19 in children with PRD, a significant majority (97%) were either asymptomatic (10%) or presented with mild symptoms (87%), with the remaining 3% presenting with moderate symptoms. No severe cases or deaths were reported.

JIA and SLE occupied the top two diseases in both PRD Center Facilities and non-PRD Center Facilities (Table 2). Asymptomatic rates were 12% in the PRD Center Facilities and 0% in the non-PRD Center Facilities. Regarding the severity of COVID-19, 98% were asymptomatic/mild, and 2% were moderate cases at the PRD Center Facilities, while 92% were asymptomatic/mild, and 8% were moderate cases at the non-PRD Center Facilities.

### 3.3. Treatment Received by Children with PRD before and during COVID-19

The relationship between pharmacological agents used to treat PRD and COVID-19 severity was analyzed by the responses to Q6 and Q7. Of the asymptomatic/mild (n = 137) and moderate (*n* = 4) cases, 77 and 3 cases were receiving glucocorticoid treatment, respectively, and the difference was not statistically significant (*p* = 0.63). Of the asymptomatic/mild (*n* = 137) and moderate (*n* = 4) cases, 100 and 4 cases were receiving immunosuppressants, respectively, and the difference was not statistically significant (*p* = 0.57). Of the asymptomatic/mild (*n* = 137) and moderate (*n* = 4) cases, 66 and 3 were using biologics, respectively, and the difference was not statistically significant (*p* = 0.36). To summarize the above, regarding the use of glucocorticoids, immunosuppressants, or biologics for the treatment of PRD prior to COVID-19, no significant difference was found between asymptomatic/mild and moderate COVID-19 in children with PRD.

Table 3 shows the responses to Q8–Q10. A total of 16 (52%) of the 31 hospitals that had patients with PRD and COVID-19 suspended the administration of or reduced the dosage of pharmacological agents for PRD because of the onset of COVID-19. The hospitals that chose to suspend a treatment were in the majority: 0 (0%) for glucocorticoids, 12 (75%) for immunosuppressants, and 13 (81%) for biologics among 16 hospitals. The hospitals that chose to reduce a treatment dosage were in the minority: zero (0%) for glucocorticoids, three (100%) for immunosuppressants, and two (67%) for biologics, among three hospitals. When children with PRD contracted COVID-19, 15 of the 31 hospitals did not suspend or reduce pharmacological agents for PRD, including glucocorticoids. Of the remaining 16 hospitals, 13 suspended any pharmacological agent for PRD, but none of those 13 hospitals suspended glucocorticoids. The remaining 3 hospitals suspended or reduced any pharmacological agent for PRD, but none of those 3 hospitals suspended or reduced glucocorticoids. Thus, glucocorticoids were not suspended or reduced (although they may have been increased) in all 31 hospitals.

## 4. Discussion

This study summarizes the clinical features of COVID-19 in children with PRD at 38 hospitals in Japan, contributing to the current knowledge in this area. There is a paucity of literature on the course of COVID-19 in children, particularly those with PRD. Researchers face a multitude of challenges when conducting studies in this population, including several children with asymptomatic COVID-19 and parental reluctance to enroll children in studies during the current pandemic [22]. It is well known that children are not miniature adults. Thus, it is essential to gather and accumulate research on PRD specifically rather than relying solely on studies of adult rheumatic diseases.

At this stage, the most important study on the clinical features of COVID-19 in patients with PRD is by Kearsley-Fleet et al., using data from the EULAR COVID-19 Registry, the CARRA Registry, and the COVID-19 Global Pediatric Rheumatology Database [23]. In this study, 607 patients with PRD from 25 countries in Europe and the United States were studied, with a female-to-male ratio of 66:34, a median age of 14 years, and three-quarters having JIA, SLE, MCTD, or vasculitis. The patient demographic in our study was similar, with a female-to-male ratio of 73:26, patients primarily aged between 11–15 years, and three-quarters having JIA, SLE, MCTD, or vasculitis (Table 2).

Kearsley-Fleet et al. reported that 93% of patients with PRD did not require hospitalization for treatment of the COVID-19 infection [23]. Our study showed a similar trend, where 97% of patients with COVID-19 were asymptomatic or had mild symptoms. In contrast to their report of two SLE deaths caused by COVID-19 [23], our study did not observe any deaths or severe cases (Table 2). Although potential differences in cohort size, our findings suggest that COVID-19 is not a significant threat to patients with PRD in Japan.

In many areas of Japan, medical expenses for children are free due to subsidy programs. Even hospitalization charges were free for children in the hospitals included in this study. If the patient or their family were worried about the illness, hospitalization could be suggested even in cases of mild symptoms. In addition, during the coronavirus pandemic, there were certain times or specific hospitals where all patients with COVID-19 were hospitalized, regardless of the severity of their symptoms. According to a nationwide survey by the Japan Pediatric Society, a cumulative total of 10,334 children < 20 years contracted COVID-19 from 2020 to 2022 [24]. Of these, the number of children treated in the outpatient, hospital, or intensive care settings was as follows: 338, 995, or 10 children < 1 year (hospitalization rate 75%); 1394, 1502, or 22 children at ages 1–4 (hospitalization rate 52%); 1828, 1273, or 19 children aged 5–9 (hospitalization rate 41%); 1423, 1002, or 10 children aged 10–14 (hospitalization rate 42%); 148, 216, or 7 children aged 15–17 (hospitalization rate 60%); and 24, 120, or 3 children aged 18–19 (hospitalization rate 82%), respectively. In this survey, 5,776 children had no treatment, 4226 used acetaminophen, 234 used glucocorticoids, 114 used remdesivir, and 0 used other pharmacological agents for COVID-19, including ritonavir-boosted nirmatrelvir. Considering that 0.7% to 3.8% of children with COVID-19 require hospitalization in the United States from 2020 to 2022 [25], the above Japanese data suggest a high hospitalization rate among children with COVID-19 in Japan. However, most children with COVID-19 were asymptomatic or mildly symptomatic in Japan [26], and the treatment was mostly involved no treatment or only acetaminophen [24]. Furthermore, the Japan Pediatric Society recommends the use of remdesivir for children with moderate to severe COVID-19 [27]. Considering that remdesivir was actually used in 114 of the 10,334 children with COVID-19, the number of moderate-to-severe cases is low from a therapeutic point of view. The unique healthcare environment in Japan, where children can receive free treatment both as outpatients and inpatients, may have contributed to the lack of severe cases and deaths among children with concurrent PRD and COVID-19.

For patients with COVID-19, our study found that 10% were asymptomatic (Table 2), while Kearsley-Fleet et al. reported in their study that 23% were asymptomatic [23]. Previous studies have found that 18% of cases in the United States [28], 24% in Germany [29], 38% in Spain [30], 13% in Turkey [19], and 21% in Portugal [31] were asymptomatic. These differences are thought to be influenced by varying definitions of “asymptomatic” and unknown COVID-19 therapeutic interventions. Additionally, there may be other influencing factors, including the environment, SARS-CoV-2 variants, patient age, PRD type, and study design. Since our study was a questionnaire survey of pediatricians/physicians, there may have been fewer asymptomatic recruits than in cohort studies. There were more asymptomatic COVID-19 cases in children with PRD at the PRD Center Facilities than at the non-PRD Center Facilities. This could be attributed to the fact that the former group receives regular follow-ups for children with PRD, increasing awareness and detection of asymptomatic cases of COVID-19.

In our study, the use of glucocorticoids, immunosuppressants, and biologics did not increase the severity of COVID-19. This finding is similar to and supports the results of a previous report [23]. However, it would be interesting to see how Japanese pediatricians/physicians coordinated pharmacological agents for PRD treatment when patients with PRD contracted COVID-19. According to the guidance provided by the American College of Rheumatology, it is recommended that patients with PRD and COVID-19 continue taking glucocorticoids at the lowest effective dose necessary to manage their PRD. However, there may be exceptions for immunosuppressants and biologics, which are generally advised to be temporarily postponed or withheld during the COVID-19 infection [32]. In our study, glucocorticoids were continued in all patients with COVID-19 (Table 3). Some hospitals have attempted to reduce the dose of immunosuppressants and biologics, but many have discontinued these medications. The treatment plan of these doctors may have reduced the severity of COVID-19. Our results suggest that, with regard to glucocorticoids, it was common for Japanese pediatricians/physicians to continue at the same (or higher) dose when treating COVID-19 cases (Table 3). On the other hand, the biologic tocilizumab is not only used for PRD treatment but also for COVID-19 treatment [33]. It was recently suggested that the Janus kinase (JAK) inhibitor baricitinib may be effective for COVID-19 treatment [34]. As a future prospect, when patients with PRD who are using biologics or JAK inhibitors contract COVID-19, it is desirable to present a new policy regarding the continued use of these drugs.

As of 31 August 2022, tocilizumab has not been approved for use in children with COVID-19 in Japan. Baricitinib, antiviral antibody drugs, and anti-SARS-CoV-2 drugs other than remdesivir and ritonavir-boosted nirmatrelvir have not been approved for use in children in Japan. Thus, they were not available as treatment options during our survey period. Our results did not find severe cases or deaths among children with PRD with moderate COVID-19, although the choice of pharmacological agents used may have influenced these outcomes. Remdesivir has been approved by the Food and Drug Administration for use in both hospitalized and nonhospitalized children [35]. In a trial evaluating remdesivir for use in nonhospitalized, high-risk patients with mild-to-moderate disease, a 3-day infusion was associated with an 87% reduction in the risk of hospitalization or death [36]. Although only eight patients aged < 18 years were included in the trial, remdesivir is recommended for children with a new or increasing oxygen need [37]. The Food and Drug Administration has issued an emergency use authorization for ritonavir-boosted nirmatrelvir for nonhospitalized children with mild-to-moderate COVID-19 who are at high risk for progression to severe disease [38]. Although the efficacy of ritonavir-boosted nirmatrelvir has only been demonstrated in adults, with an 89% relative risk reduction compared to placebo [39], it is recommended for children who are at high risk of progressing to severe disease [40]. However, drug interactions must be carefully considered when administering ritonavir-boosted nirmatrelvir, which may limit its use in certain high-risk populations, such as immunocompromised children [37].

Compared to previous reports on COVID-19 with PRD, our study is novel because it included children infected who are during the Omicron variant epidemic. Japan experienced the pandemic of Omicron subvariant BA.1/BA.2 from January to June 2022. However, after the emergence of BA.5 in July 2022, the number of children hospitalized with COVID-19 increased dramatically. Compared with the Omicron subvariant BA.1/BA.2, BA.5 proved to be more detrimental, and the incidence of neurological complications requiring hospitalization increased [41]. Our study likely included numerous children with PRD who were infected not only with the BA.1/BA.2 subvariant but also with the BA.5 Omicron subvariant. The results suggest that children with PRD who contract COVID-19 due to the Omicron variant are unlikely to experience severe disease.

This study had several limitations. First, there was a low response rate, and some cases were omitted for answers about COVID-19 severity in this survey. The definition of COVID-19 severity used in this study was not previously reported but is original. Furthermore, in our study, we did not investigate the diagnosis method, symptoms, treatment of COVID-19, the severity of COVID-19 for each child with PRD, relapse of PRD due to COVID-19, reasons for the suspension, reduction, no suspension, or no reduction in pharmacological agents used to treat PRD during COVID-19 infection, or the clinical features of infection with each SARS-CoV-2 variant. To address these limitations, a future study will be needed.

## 5. Conclusions

This is the first survey on the clinical features of concurrent COVID-19 and PRD in the Japanese population. In total, 97% of the children with PRD had asymptomatic or mild COVID-19. No severe cases or deaths were reported. Thus, COVID-19, including infection with the Omicron variant, is not a threat to children with PRD in Japan. When children with PRD develop COVID-19, glucocorticoids should be continued, whereas immunosuppressants and biologics may be suspended.

## Figures and Tables

**Table 1 viruses-15-01205-t001:** Questionnaire used for the survey of the PRAJ members.

Q1. Were there any children with a pediatric rheumatic disease (PRD) with COVID-19 in your hospital?
Q2. Kindly specify the sex-based count of children with PRD who have contracted COVID-19.
Q3. Kindly specify the age-based count of children with PRD who have contracted COVID-19.
Q4. Please indicate the type of PRD and the number of children with COVID-19.
Q5. Please provide the number of children with PRD for each COVID-19 severity (asymptomatic, mild, moderate, severe, or death).
Q6. Please provide the number of children with PRD receiving each pharmacological agent for PRD (glucocorticoids, immunosuppressants, or biologics) before asymptomatic or mild COVID-19.
Q7. Please provide the number of children with PRD receiving each pharmacological agent for PRD (glucocorticoids, immunosuppressants, or biologics) before moderate or severe COVID-19.
Q8. Have you suspended the administration or reduced the dosage of the pharmacological agents for PRD (glucocorticoids, immunosuppressants, or biologics) because of the presence of COVID-19?
Q9. If you answered yes to Q8, which pharmacological agents for PRD (glucocorticoids, immunosuppressants, or biologics) did you suspend?
Q10. If you answered yes to Q8, which pharmacological agents for PRD (glucocorticoids, immunosuppressants, or biologics) did you reduce the dosage of?

Abbreviations: COVID-19, coronavirus disease 2019; PRAJ, Pediatric Rheumatology Association of Japan; PRD, pediatric rheumatic disease.

**Table 2 viruses-15-01205-t002:** Demographic characteristics and severity of COVID-19 in children with PRD.

	In Total 38 Hospitals *n* (%)	In 22 PRD Center Facilities *n* (%)	In 16 Non-PRD Center Facilities *n* (%)
Sex	156 (100)	131 (100)	25 (100)
Female	114 (73.1)	97 (74.1)	17 (68.0)
Male	42 (26.9)	34 (26.0)	8 (32.0)
Age (years)	156 (100)	131 (100)	25 (100)
0–5	14 (9.0)	13 (9.9)	1 (4.0)
6–10	36 (23.1)	30 (22.9)	6 (24.0)
11–15	73 (46.8)	59 (45.0)	14 (56.0)
16–17	33 (21.2)	29 (22.1)	4 (16.0)
Type of PRD	156 (100)	131 (100)	25 (100)
JIA *	81 (51.9)	69 (52.7)	12 (48.0)
Polyarticular	33 (21.2)	31 (23.7)	2 (8.0)
Systemic	21 (13.5)	17 (13.0)	4 (16.0)
Oligoarticular	18 (11.5)	13 (9.9)	5 (20.0)
Others	9 (5.8)	8 (6.1)	1 (4.0)
SLE	37 (23.7)	30 (22.9)	7 (28.0)
Juvenile dermatomyositis	7 (4.5)	5 (3.8)	2 (8.0)
Scleroderma	6 (4.0)	6 (4.6)	0 (0)
Takayasu arteritis	5 (3.2)	5 (3.8)	0 (0)
Chronic recurrent multifocal osteomyelitis	4 (2.6)	2 (1.5)	2 (8.0)
Sjögren’s syndrome	3 (2.0)	3 (2.3)	0 (0)
Antiphospholipid antibody syndrome	2 (1.3)	2 (1.5)	0 (0)
Behçet’s disease	2 (1.3)	2 (1.5)	0 (0)
MCTD	2 (1.3)	2 (1.5)	0 (0)
Polyarteritis nodosa	2 (1.3)	1 (0.8)	1 (4.0)
Arthritis associated with Crohn’s disease	1 (0.6)	1 (0.8)	0 (0)
Juvenile polymyositis	1 (0.6)	1 (0.8)	0 (0)
Overlap syndrome	1 (0.6)	1 (0.8)	0 (0)
Tubulointerstitial nephritis and uveitis syndrome	1 (0.6)	0 (0)	1 (4.0)
Uveitis	1 (0.6)	1 (0.8)	0 (0)
Severity of COVID-19 **	141 ^§^ (100)	116 ^§^ (100)	25 (100)
Asymptomatic	14 (9.9)	14 (12.1)	0 (0)
Mild	123 (87.2)	100 (86.2)	23 (92.0)
Moderate	4 (2.8)	2 (1.7)	2 (8.0)
Severe	0 (0)	0 (0)	0 (0)
Death	0 (0)	0 (0)	0 (0)

Abbreviations: COVID-19, coronavirus disease 2019; JIA, juvenile idiopathic arthritis; MCTD, mixed connective tissue disease; PRD, pediatric rheumatic disease; SLE, systemic lupus erythematosus. * International League of Associations for Rheumatology classification criteria [21]. ** Mild cases were defined as those that improved without using pharmacological agents for COVID-19; moderate cases were defined as those that used pharmacological agents for COVID-19 but did not require intensive care unit management; and severe cases were defined as those that required intensive care unit management. ^§^ There were 15 cases in which the severity of COVID-19 was omitted from the answer in this survey.

**Table 3 viruses-15-01205-t003:** Adjustment of pharmacological agents used to treat PRD during COVID-19 infection by pediatricians/physicians.

	Hospitals with Children with PRD and COVID-19 (*n* = 31)				
	Not Suspended/ Not Reduced (*n* = 15)	Suspended (*n* = 13)	Suspended/Reduced (*n* = 3)		Reduced (*n* = 0)
			Suspended (*n* = 3)	Reduced (*n* = 3)	
Glucocorticoids (n)	N/A	0	0	0	N/A
Immunosuppressants (n)	N/A	9	3	3	N/A
Biologics (n)	N/A	11	2	2	N/A

Abbreviations: COVID-19, coronavirus disease 2019; N/A, not applicable; PRD, pediatric rheumatic disease.

## Data Availability

Documents related to the questionnaire survey are available from the corresponding author upon reasonable request.

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
