# Peer review of "Clinical Features of COVID-19 in Pediatric Rheumatic Diseases: 2020–2022 Survey of the Pediatric Rheumatology Association of Japan"

_viruses, 2023, doi:10.3390/v15051205_

Round 1
Reviewer 1 Report
Please find the comments in the attached file

Minor English improvements are needed
Reviewer 2 Report
The authors of this manuscript present a summary of the management of children with rheumatology diseases in Japan who also contracted COVID-19 infection. The study included 165 children. Of importance, none died from COVID infection. The manuscript is well written and the subject matter is of importance, especially because there are no prior data from Japanese children. A few comments are listed below.
1.Discussion, line 153. Add a few sentences about the healthcare system for children in Japan. What is the cost to Japanese parents to insure their children? What is the cost to Japanese parents if their children are admitted to hospital? Differences in healthcare systems and costs of insurance may also be an explanation for differences in deaths of children with serious diseases in different countries. Please add this information into the Discussion.
2.Limitations, line 178. Add the low response rate of 12% as a possible limitation.
Round 2
